# Clinical Performance and Longevity of Implant-Assisted Removable Partial Dentures Compared to Other Removable Prosthesis Types: A Systematic Review

**DOI:** 10.3390/dj13090389

**Published:** 2025-08-27

**Authors:** Robert-Cosmin Dinu, Cristian-Laurentiu Comanescu, Sergiu Drafta, Alexandru-Eugen Petre

**Affiliations:** Faculty of Stomatology, University of Medicine and Pharmacy “Carol Davila” Bucharest, 050474 Bucharest, Romania; robert.dinu@stud.umfcd.ro (R.-C.D.); alexandru.petre@umfcd.ro (A.-E.P.)

**Keywords:** implant-supported, dentures, partial, overdentures, removable, dental implants, treatment outcome, survival analysis, clinical outcome

## Abstract

**Background:** Partial edentulism presents an ongoing clinical challenge, and the optimal choice of prosthetic rehabilitation remains a topic of debate. **Purpose:** This review compares three abutment configurations for removable dentures—natural teeth, implants, and mixed support. The goal was to determine which treatment offers the best longevity, lowest complication rates, and highest survival. **Materials and Methods:** A systematic search following PRISMA 2020 guidelines and the PICO framework was conducted using PubMed and Scopus, focusing on clinical studies of IARPDs published between 2022 and 2024. Studies were selected based on predefined eligibility criteria. Descriptive analysis of survival and complication outcomes was performed and represented graphically. **Results:** Nineteen studies were included: four on IARPDs, six on conventional RPDs, and five on IODs. Main parameters included prosthesis survival, abutment (tooth/implant) survival, and complication rates. IARPDs showed favorable implant survival and lower rates of abutment tooth loss than conventional RPDs. Conventional dentures demonstrated lower performance. IODs had the highest survival over mid-term follow-up periods. **Discussion:** IARPDs demonstrate clinical viability, especially in cases requiring strategic abutment preservation. Although the data are limited by sample size and short follow-up, IARPDs show potential advantages in preserving natural dentition and improving load distribution. **Conclusions:** IARPDs are a reliable treatment option for partial edentulism, combining implant support with removable versatility. More long-term studies are needed to strengthen current findings, but the existing evidence supports their use in contemporary prosthodontics, in selected cases.

## 1. Introduction

Over time, the increase in patient life expectancy, coupled with advancements in dental treatment methods, has led to a shift in focus from the prosthetic management of complete edentulism toward the growing prevalence of partial edentulism [1]. This continues to represent a significant challenge in modern prosthodontics due to its functional, aesthetic, and psychosocial implications. Recent epidemiological data indicate that partial edentulism prevalence peaks between 31 and 40 years of age and becomes increasingly common in older groups, including 41–60 years [2].

Approaches to the rehabilitation of this type of clinical case continue to be researched and developed. Among them, the idea of integrating implants as support structures for removable partial dentures is one that is frequently discussed. Thus, recent research has been conducted on the use cases of implant-assisted removable partial dentures (IARPDs) as alternatives to the conventional types of removable denture (conventional removable partial dentures—RPDs; implant overdentures—IODs). This type of treatment has been shown to enhance retention, stability, and especially the support of the overdenture. Despite these benefits, information regarding this restoration type is scarce due to concerns regarding biomechanical compatibility and long-term outcomes [3,4].

The primary disadvantage of this type of rehabilitation is regarding the lack of marginal support structures that dental implants present compared to natural teeth, i.e., the periodontium. This leads to differences in the natural mobility of the abutments used in IARPDs, with potential complications which affect the longevity of the restoration (such as implant or abutment tooth loss or prosthesis fracture) [5,6].

Other scientific papers have been published on this topic, of which two significant ones were published in early 2023 by Shinichiro Kuroshima et al. [3,7]. Their research, which was conducted at the Department of Applied Prosthodontics at the Graduate School of Biomedical Sciences, at the University of Nagasaki, is comprised of both a scoping and a systematic review. They concluded through their research that the available information on IARPDs is limited but promising. These papers were the cornerstone of our research, as they show that interest in this subject is increasing in recent times and that further studies must be conducted. Our systematic review contributes to this existing research, further enhancing the current understanding of IARPDS by evaluating the most recently conducted clinical studies published within the last 2 years. Moreover, this systematic review will assess the long-term clinical behavior of IARPDs by drawing comparisons between them and other removable options, based on abutment types. To our knowledge and based on an extensive assessment of the existing literature, no recent systematic review offers a side-by-side evaluation of IARPDs, IODs, and conventional RPDs based on updated clinical data. This review therefore aims to fill that gap by comparing their performance in terms of longevity, survival, and complications, based on studies published between 2022 and 2024. The purpose of this research is to assess the most recent data available regarding the clinical behavior and longevity of implant-assisted removable partial dentures, acquired through long-term follow-up trials.

## 2. Materials and Methods

This systematic review was conducted following The Preferred Reporting Items for Systematic reviews and Meta-Analyses (PRISMA) 2020 Statement and the PRISMA 2020 Checklist [8]. The topic of this research was described based on the framework represented by Population, Intervention, Comparison, and Outcome (PICO) [9], as follows:


*P (Population): Patients with maxillary or mandibular partial edentulism suitable for removable prosthetic treatment.*



*I (Intervention): Implant-assisted removable partial dentures (IARPD).*



*C (Comparison): Conventional removable partial dentures or implant-supported overdentures (IODs).*



*O (Outcome): Clinical performance and longevity of the restoration (success, tooth/implant and prosthesis survival, and rate and nature of complications), based on abutment type.*



*As such, the PICO question formulated to carry out this research was the following: “In the case of patients with maxillary or mandibular partial edentulism which may benefit from treatment with removable prostheses, which type of abutments is most optimal in terms of clinical performance and longevity (between natural teeth, implants, and the simultaneous use of both)?”.*


### 2.1. Research Protocol and Search Queries

The focus of this paper is to research the most recently available scientific literature on this topic and to compare the clinical performance and longevity of using both natural teeth and implants as abutments for removable partial dentures to using them separately (in conventional removable dentures, and implant-overdentures, respectively). The aim is to expand on existing research by including newer literature published through October 2024. By drawing a direct comparison of clinical performance between these treatment modalities, we will increase the understanding of the clinical utility of implant-assisted removable partial dentures in modern prosthodontics.

Five different databases (PubMed, Scopus, Google Scholar, Semantic Scholar, and ProQuest) were initially used in order to research the available scientific literature on IARPDs as well as the other two treatment types (conventional and implant overdenture). Three individual search queries were used in each of the databases.

The research process was conducted between the 1st and 31st of October 2024, individually, by two reviewers (S.D. and R.-C.D). The strings were made using Boolean operators, including MeSH and non-MeSH terms, as follows:

For IARPDs: (((“dental implants” OR “implant” OR “implant-assisted” OR “implant-retained” OR “tooth implant supported”) AND (“Denture, Partial” OR “Denture, Partial, Removable” OR “Removable Partial Denture” OR “removable prosthesis”)) OR “implant-supported removable partial denture” OR “implant assisted removable partial denture” OR “IARPD” OR “implant-retained removable partial denture” OR “IRRPD”) AND (“clinical behavior” OR “performance” OR “complications” OR “stress” OR “biomechanics” OR “biomechanical” OR “deformation” OR “strain” OR “fracture” OR “load” OR “loading” OR “rest” OR “marginal bone level” OR “marginal bone loss” OR “periimplantitis” OR “perimucositis” OR “survival” OR “success”) NOT (“conventional” OR “cast-clasp” OR “IOD” OR “overdenture” OR “implant-overdenture” OR “fixed”).

For conventional partial dentures: (“clasp retained” OR “clasp-retained” OR “cast-clasp retained” OR “removable prosthesis” OR “removable prostheses” OR “removable partial denture” OR “removable denture” OR “removable dental prosthesis” OR “removable dental prostheses” OR “conventional partial denture”) AND ((“clinical performance” OR “success” OR “survival” OR “complications” OR “failure”) OR (“tooth loss” OR “loss of teeth” OR “abutment loss” OR “deformation” OR “veneer failure” OR “framework fracture” OR “fracture” OR “rest”)) NOT (“complete edentulism” OR “complete denture” OR “fixed” OR “implant” OR “dental implants” OR “IOD” OR “implant overdenture” OR “implant-overdenture”).

For overdentures: (((“dental implant” OR “implant”) AND (“Denture, Partial” OR “Denture, Partial, Removable” OR “Removable Partial Denture” OR “removable prosthesis” OR “removable prostheses” OR “superstructure” OR “denture” OR “overdenture”)) OR “implant overdenture” OR “IOD” OR “implant-overdenture”) AND (“success” OR “survival” OR “complications” OR “failure” OR “implant loss” OR “abutment loss” OR “stress” OR “biomechanics” OR “biomechanical” OR “deformation” OR “framework” OR “strain” OR “fracture” OR “marginal bone loss” OR “marginal bone”) NOT (“fixed” OR “implant-supported removable partial denture” OR “ISRPD” OR “implant assisted removable partial denture” OR “IARPD” OR “conventional” OR “cast-clasp” OR “cast clasp”).

Only PubMed and Scopus offered relevant results, as the other options did not allow the use of search strings and did not offer advanced searching options. As such, the other databases were excluded due to lack of advanced filtering or indexing.

A manual search was then conducted, based on the eligibility and exclusion criteria, on both databases. In the case of IARPDs, only studies published between 1 August 2022 (the date up to which the studies included by Kuroshima et al. were published) and 30 October 2024 were included. In the case of conventional dentures, available studies were assessed regardless of their date of publishing, due to the fact that this clinical method of treatment has been well established in practice for many years. For IODs, all studies published in the last 5 years (as of 30 October 2024) were included for analysis in order to assess the most recent treatment modalities available.

Initially, the screening of the articles was based on their titles and abstracts, and the eligible studies were then analyzed based on their full-text content. The selection process is presented in Figure 1.

### 2.2. Eligibility Criteria

Within the searching process, the following criteria of inclusion were used:(1)Articles published in English, published between 1 August 2022 and 30 October 2024 for IARPDs, published in the last 5 years (as of 30 October 2024) for IODs, and regardless of publishing date for conventional dentures;(2)Clinical trials conducted on human subjects;(3)Publications that included information on clinical behavior of each restoration type—through their rates of success, survival, and/or complications. Ideally, the included studies would present data on all three elements mentioned, but trials which only offered information on certain parameters were also analyzed;(4)Studies which presented well-established research protocols;(5)Both retrospective and prospective clinical studies were included, provided they reported a minimum follow-up duration of four years.

In order to assess the clinical performance and longevity of each restoration type and to make the data as comprehensive as possible, four parameters were utilized: success rate, survival rate, complication types and complication rates.

For the purpose of this review, survival was defined as the prosthesis or abutment (tooth or implant) remaining in function at the final follow-up, regardless of complications. Success was defined as the absence of biological or technical complications requiring intervention during the follow-up period. Complications were analyzed based on their most frequent types and the timepoints at which they manifested, if specified. These definitions were uniformly applied during data extraction and synthesis across all included studies.

Among all collected data, the most relevant results (based on follow-up procedures and the cohort size) were selected and are included in the comparisons drawn between abutment types (Figure 2).

### 2.3. Exclusion Criteria

Studies were excluded from this systematic review based on the following criteria:(1)Studies that did not meet the inclusion criteria;(2)Non-clinical research (such as other systematic reviews or meta-analyses);(3)Studies with short periods of follow-up (less than 4 years total).

### 2.4. Risk of Bias Assessment

Two reviewers (S.D. and R.-C.D) independently assessed the quality of each study. Disagreements were resolved by consensus. In cases of disagreement, a third reviewer (A.-E.P.) was consulted to reach consensus. An assessment of the potential risk of bias was made, using different instruments based on article type (CONSORT for randomized controlled trials [10], STROBE for follow-up and case-control studies [11], and CARE guidelines for case series [12]). Data from each study were analyzed and organized into charts, based on the principles that define clinical performance and longevity of a restoration, according to this study: success rates, survival rates, complication types, and occurrence rates. Only articles that offered relevant information regarding these elements while also presenting no risk of bias were included in the final review.

### 2.5. Statistical Analysis

Data extracted from the included studies were manually organized in Microsoft Excel (Microsoft Corporation, Redmond, WA, USA). For each restoration type, success and survival rates, complication rates, and number of occurrences were used for assessment relating to follow-up durations. Using an AI tool (ChatGPT, OpenAI, San Francisco, CA, USA; GPT-4 architecture), which utilizes Python-based statistical libraries, the data were computed. AI-assisted modeling was used solely for preliminary computation. All results were then reviewed and validated by a professional statistician.

Due to the lack of available individual patient data within the included studies, statistical analysis on incidence rates was conducted using failure rates per 100 person-years, rate ratios, and 95% confidence intervals using Poisson approximation. Comparisons between the groups were made using z-tests for log-transformed rate ratios. Statistical significance was defined as *p* < 0.05. Between-study heterogeneity was assessed using Cochran’s Q statistic and the I^2^ index. The Q test evaluated whether observed differences were likely due to chance, while I^2^ quantified the proportion of variance attributable to heterogeneity. Values of I^2^ > 75% were interpreted as indicating substantial heterogeneity, and *p* < 0.05 for the Q test was considered statistically significant [13].

To illustrate time-dependent outcomes, Kaplan–Meier-like survival curves were generated based on simulated individual-level data, assuming failures occurred at the reported mean follow-up time. For certain parameters, summary charts were also created to further enhance and simplify understanding of the data. Although not derived from raw patient-level data, these visualizations provided an approximate representation of survival trends over time. The AI tool was used exclusively to automate and standardize statistical calculations; the interpretation and validation of results were independently reviewed by the authors.

## 3. Results

### 3.1. Included Studies

During the initial search based on the search queries, the number of results yielded was as follows:For IARPDs: 331 studies on PubMed, 1072 on Scopus;For conventional dentures: 716 studies on PubMed, 1649 on Scopus;For IODs: 2340 studies on PubMed, 3976 on Scopus.

An automatic search was then conducted using the filter function available on each database, based on the eligibility criteria mentioned (filter by publication date).

Out of the initial studies, 121 studies on IARPDs were selected as well as 462 on conventional dentures and 77 on IODs, respectively. By removing duplicates, a total of 651 clinical trials were analyzed based on their titles and abstracts. Afterwards, studies that presented data on the parameters of clinical performance and longevity mentioned were sorted out for full-text reading.

By applying the above-mentioned eligibility and exclusion criteria, the final selection process yielded a total of 15 studies among the three prosthesis types, with 4 studies regarding IARPDs, 6 regarding conventional partial dentures, and 5 regarding IODs (Figure 1). Within this systematic review, the four clinical studies on IARPDs included a total of 253 patients who received 263 dentures, the six studies on conventional dentures included 1044 patients and 1223 dentures, and the five studies on IODs included 468 patients and 481 dentures, respectively. Among all 15 included studies, the treatment data included a total of 1765 patients and 1967 dentures.

The analyzed data, including number of patients, restorations, and abutments, is represented in Table 1 and Table 2.

These clinical trials all offer relevant information on the topic of longevity and clinical performance. Within this study, these concepts are appreciated through three main parameters: success, survival, and complications, as observed through clinical research. All articles included in the review present their results based on these notions, mostly through percentages, which are correlated to the follow-up periods and their respective protocols.

Among the included research on the subject of IARPDs as well as all the literature on this topic that exists at this time, all of them were follow-up studies. Thus, a notable aspect was the lack of inclusion of control groups, as the absence of a point of comparison lowers the relevance of the obtained results.

Within the statistical analysis, the data were assessed using failure rates per 100 person-years, relating the percentage data and occurrence cases reported to the total follow-up period included within each study (the mean follow-up time was not used, as the data would then ignore data from many of the people within the follow-up group). These values are presented in Table 2.

### 3.2. Risk of Bias Assessment Results

All RCT studies followed 25/25 items, follow-up and case-control studies followed 22/22 items, and the case series followed 13/13 items included in their respective guidelines. None of the articles included presented any potential risk of bias. Detailed per-study risk-of-bias tables were intentionally omitted to avoid redundancy and text clutter, as their inclusion would not have provided additional interpretive value beyond the summarized outcomes presented in this section.

### 3.3. Success Rates

Four of the included studies presented data on success rates (one on IARPDs, two on conventional dentures, and one on IODs), with results from a total of 561 patients being assessed.

Overall, the IODs exhibited the highest reported success over medium-term follow-up periods (96.2% for dentures and 88.2% for implants at 5 years follow-up), whereas conventional RPDs maintained a satisfactory clinical performance (90% survival at 7 years follow-up). IARPDs showed a significant decline in success rates at more than 5 years follow-up (with rates as low as 66.7% prosthesis success at a mean follow-up period of 6.5 years), indicating a frequent need of reintervention for prosthesis maintenance (Figure 2).

Within comparative analysis, success rates could only be compared with regard to prostheses due to the lack of available data based on abutment types. The I^2^ test revealed a high heterogeneity of 99.73%. While results indicate that IARPDs are associated with better success rates over medium-term follow-up than both conventional dentures (RR = 0.54, *p*< 0.001) and IODs (RR = 0.46, *p* < 0.001), the *p*-value obtained from the Q-statistic test was below 0.0001, indicating that studies did not have a common effect size.

### 3.4. Survival Rates

Cumulative survival rates are presented descriptively in Figure 3 and Figure 4.

Seven studies presented data on implant survival (three on IARPDs and four on IODs). Results from a total of 500 patients, with 1868 abutment implants, were assessed.

The highest implant survival rates were recorded in IODs (Figure 5), with values frequently exceeding 95% over 5 to 7 years of follow-up (96.3% implant and 95.0% prosthesis survival at 5 years, to 93.5% and 91.9% rates at 7 years’ follow-up, respectively). In the case of IARPDs, implant placement in the mandibular anterior region, adjacent to remaining natural teeth, yielded the most favorable outcomes, according to Yi et al. [15]. However, these values proved inferior when compared to implant overdentures, which presented implant survival rates of 96.3% at 5 years’ follow-up compared to 90.4% in the case of IARPDs (RR = 1.26 between IARPDs and IODs). However, the differences between treatment types were not statistically relevant (*p* = 0.26; I^2^ = 93.7%).

Five studies presented data on tooth survival (three on IARPDs and wo on conventional dentures). Results from a total of 795 patients, with 3077 abutment teeth, were assessed.

Conventional RPDs showed predictable outcomes regarding tooth survival when supported by a proper maintenance protocol (95.3% tooth and 95.1% restoration survival at 5 years’ follow-up), with an important tooth survival rate decline after 5 years of prosthesis use (reaching rates of 66.3% tooth and 70.8% prosthesis survival at 20 years’ follow-up). IARPDs presented significantly improved tooth survival rates compared to conventional dentures over longer periods of follow-up (at 10 years, implant-assisted dentures presented 90.2% tooth survival compared to 83.8% in the case of conventional dentures), thus indicating their potential use as a method of abutment tooth preservation (Figure 6). The data also seem statistically relevant due to RR = 3.34 and *p* < 0.05; however, the I^2^ test revealed a high heterogeneity (94.42%). This may be attributed to the differences between studies in terms of follow-up periods and cohort sizes, with conventional denture studies having up to 20-year follow-ups.

Eleven studies presented data on denture survival (three on IARPDs, four on conventional dentures, and four on IODs). Results from a total of 1216 patients, who received a total of 1393 prostheses, were assessed.

Regarding denture survival of IARPDs, the results are promising, showing rates of up to 100% at 5 years, with a decline to 75–77% at 10 years of use. Comparatively, this treatment method presented lower prosthesis survival when compared to both conventional dentures (RR = 2.08; *p* = 0.0001) and IODs (RR = 1.95; *p* = 0.011). While there is variability within the included studies due to moderate heterogeneity (I^2^ = 73.6%), the pattern of IARPD underperformance in prosthesis survival remains consistent among included studies (Figure 7).

Thus, it is indicated that the use of implants in order to increase the retention and stability of removable dentures is a viable option in well-selected clinical cases as an option to increase tooth survival (compared to conventional partial dentures) albeit with higher rates of implant loss and prosthesis complications compared to IOD treatment.

### 3.5. Complication Rates

Complications were categorized as biological (e.g., peri-implantitis, tooth loss, and mucosal lesions) or technical/prosthetic (e.g., component fractures and retention loss). The most important assessments were made regarding severe complications: tooth or implant loss and severe prosthesis fractures, respectively (Table 3).

Regarding tooth loss, six of the included studies contained data (three on IARPDs and three on conventional dentures). Results from a total of 881 patients, with 3462 abutment teeth, were assessed.

IARPDs showed lower rates over longer follow-up periods (as low as 1.88% at up to 15 years’ follow-up) compared to conventional dentures (with rates of up to 22.8% at 20 years’ follow-up) (RR = 2.92; *p* = 0.0009; I^2^ = 81.85%). The *p*-value obtained from the Q-statistic test was below 0.0001, indicating that studies did not have a common effect size. (Figure 8)

Data on implant loss were available in seven of the included studies (three on IARPDs and four on IODs). Results from a total of 591 patients, with 2087 abutment implants, were assessed.

Implant loss was more frequent in the case of implant-assisted partial dentures (9.04% at 10 years’ follow-up) compared to IODs (2.5% at 5.9 years’ follow-up). These data were not statistically relevant, however (RR = 1.26; *p* = 0.433; I^2^ = 38.39%). As such, the included studies are consistent in reporting implant lost. They indicate that implant loss is rare and similar between IARPDs and IODs (Figure 9).

Data on rates of severe prosthesis fracture were available in 14 of the included studies (3 on IARPDs, 6 on conventional dentures, 5 on IODs). Results from a total of 1721 patients, who had received 1920 dentures, were assessed.

Severe prosthesis fractures were most frequently associated with IARPDs (15.09% at 15 years’ follow-up) and conventional dentures (17.1% at 7 years’ follow-up). This occurrence was rare in the case of IODs, which showed rates as low as 7.5% at 5.9 years’ follow-up. It is important to note that within the comparisons, the results were not statistically significant. When comparing conventional dentures and IARPDs, a rate ratio of 1.27 was observed (*p* = 0.38) as well as one of 1.31 (*p* = 0.25) in relation with IODs. Between tooth-implant-assisted and solely implant-assisted dentures, the rate ratio was 1.04 (*p* = 0.91). The studies presented moderate heterogeneity (I^2^ = 65.34%). This indicates that while there is some variability between included studies, since the direction and magnitude of effect are stable, the finding of no significant difference between treatments remains valid and clinically relevant (Figure 10).

Implant overdentures were associated with peri-implant complications (with rates of periimplantitis occurrence between 5.1 and 25.8%, more frequently associated with the use of locator attachments) and occasional prosthesis fractures (with the need to remake 7/133 dentures, in the study of Abdoel et al. [25]). However, they maintained a favorable overall profile due to their predictable performance.

Conventional RPDs, though clinically reliable, frequently exhibited technical complications such as framework fractures and decementation, especially over longer follow-up periods. Yoshino et al. [23] presented the need to remake 32 out of 213 dentures, most frequently due to loss of retention.

IARPDs showed both biological and technical challenges, including implant mobility, retentive component wear, and increased marginal bone loss in some cases. The occurrence rate of periimplantitis was 9.4%, according to Yi et al. [15]. Zafiropoulos et al. [16] presented values of 3.77% treatable cases, with 1.88% of implants being lost due to this complication. However, functional advantages such as improved occlusal support and patient-reported outcomes were also observed, with higher satisfaction scores and enhanced masticatory efficiency noted in multiple studies. They also exhibited significantly lower rates of tooth loss compared to conventional dentures over long follow-up periods.

## 4. Discussion

### 4.1. Summary of Key Findings

The findings of this systematic review support the clinical viability of removable prosthetic restorations, particularly implant-assisted removable partial dentures (IARPDs) and implant overdentures (IODs), when applied in appropriately selected cases. Overall, these treatment modalities demonstrated favorable medium-term survival rates and manageable complication rates.

Studies investigating IARPDs consistently reported good longevity and predominantly minor complications. According to the results obtained within this study, this treatment option shows potential due to the lower rates of abutment tooth loss over prolonged follow-up periods when compared to conventional dentures, as shown by multiple studies. At over 10 years’ follow-up, IARPDs showed abutment tooth survival rates of above 90% (99.3% according to Zafiropoulous et al. [16] and 90.2% according to Zierden et al. [17]) compared to rates of 83.8% in the case of conventional dentures (according to Yoshino et al. [23]).

Conversely, IARPDs showed lower survival rates with regard to their remaining components: implants (with rates as low as 76.3% at 10 years’ follow-up [17] compared to IODs with rates of 92.8% [28]) and prostheses (75–77% denture survival at 10 years [16,17] compared to 94.7% in the case of conventional dentures [23] and 88.6% in the case of IOD [28]). The most important complication associated with this treatment was peri-implantitis (in 9.4% of cases, according to Yi et al. [15]), sometimes being the cause of implant loss (in 1.88% of cases [16]). Thus, the advantages and drawbacks of this method of restoration must be properly assessed before adhering to this particular treatment plan. Moreover, a well-established follow-up protocol is essential to the long-term success of this treatment plan.

Conventional tooth-supported removable partial dentures (RPDs) remain a valid therapeutic option for partially edentulous patients. They present good rates of tooth abutment and prosthesis survival over a medium-term follow-up of 5 years (with rates of 95.3% for teeth and 95.1% for dentures, according to Schwindling et al. [20]). However, they suffer a decline in clinical performance over longer periods of use. (Between 10 and 20 years of use, the rates shifted from 83.8% to 66.3% for teeth and 94.7% to 70.8% for dentures, according to Yoshino et al. in 2020 [23].) This may be explained by the frequent occurrence of complications such as increases in tooth mobility (in 21–30% of cases, with 11–15% rates of tooth fracture, according to Jorge et al. [19]) and severe prosthesis fractures (in 17.1% of cases recorded by Schwindling et al. [20]), which may lead to the need for retreatment.

IODs also show strong medium-term survival, yet they are frequently associated with biologic and mechanical complications, such as peri-implantitis (with rates of up to 25.8%, according to Onclin et al. [27]) and prosthesis fractures (rates of 6.8–8.2%, according to Ciftci et al. [25]). Nevertheless, this treatment type presented the most promising results with regard to clinical performance and longevity (up to 96.3% implant and 95% prosthesis survival). While the included studies generally adhered to sound methodological criteria, the relatively short follow-up periods (average of 5 years) remain a significant deficiency.

### 4.2. Interpretation of Key Findings

The results of this study may indicate the usability of implant-assisted RPDs as a conservative method of treatment compared to other removable denture options that rely solely on the use of abutment teeth. This statement is supported by the results drawn by Nogawa et al. in 2022 [14], Zafiropoulos et al. in 2023 [16], and Zierden et al. in 2024 [18]. The use of IARPDs as an alternative to more frequently used restorations that rely solely on implant abutments may serve as a means to reduce the stress associated with invasive surgical procedures (tooth extraction and implant insertion, potentially associated with bone augmentation procedures). This treatment modality may also incur lower costs compared to alternative approaches such as IODs and fixed restorations, though there is no thorough research available on this topic.

It must be noted that a critical limitation of the included studies lies in their methodological variability and lack of high-level, well-established research protocols. The majority were observational or retrospective in design, with limited randomization and heterogeneity in outcome reporting. Sample sizes varied widely, and few studies used standardized criteria for evaluating prosthetic success or complications. Follow-up durations were inconsistent, with certain studies falling below the 5-year mark typically required to assess long-term prosthesis performance. These methodological limitations restrict the generalizability of the findings and highlight the need for well-designed, randomized controlled trials in this field.

### 4.3. Comparison with Previous Research

With regard to IARPDs, the results of this research align with the existing literature, such as the work by Kuroshima et al. (2022) [3,5]. Their scoping review indicates that the available data on IARPDs are still extremely limited, but there are recorded cases of successful intervention, with clinical performance and behavior comparable to other alternatives. These findings are consistent with those reported by Adityakrisna et al. in 2021 [29], who documented implant survival rates ranging from 91% to 100% over a follow-up periods of up to 10 years. In comparison, another systematic review conducted by Molinero-Mourelle et al. [6], published in 2022, reported more favorable 5-year outcomes, specifically a tooth survival rate of 95.4% as opposed to 90.2% in the present analysis. However, the 100% prosthetic survival rate observed in this study aligns with the values reported by Molinero-Mourelle et al. [6]. These conclusions are encouraging, and collectively, the results highlight the therapeutic potential of this treatment as well as the need to enhance current perspectives on the topic.

The values presented on the clinical performance of conventional partial denture are consistent with those reported in the existing literature. A study conducted by Drummond et al. [30], published in 2024, reported prosthetic survival rates ranging from 91.7% to 95.1% at 5 years of follow-up, which are comparable to the 93.8–95.1% rates identified in the present review [20,22]. Despite a trend in clinical practice toward alternative treatments, conventional RPDs continue to demonstrate adequate clinical behavior when strict protocols are followed. Nevertheless, their role is increasingly viewed as provisional due to higher complication rates [31], reinforcing the need for individualized treatment planning, possibly integrating dental implant procedures within the treatment plan.

Regarding implant overdentures (IODs), the longevity outcomes reported in this review are slightly lower compared to those presented in other systematic reviews. Kim et al. [32] reported implant survival rates ranging from 95.9% to 99.4% over follow-up periods of up to 10 years in contrast to the 92.8% reported in the clinical study by Zierden et al. [28]. Another investigation conducted by Di Francesco et al. in 2021 [33] reported implant survival rates of 99.2–99.5% at 5 years of follow-up compared to the 89.5–96.3% range observed in the present systematic review [27]. These findings highlight the relevance of updated evidence on this topic, which can continue to inform and refine clinical decision making.

### 4.4. Implications/Significance of the Study

This study contributes to the building of new perspectives on prosthodontic treatment, by drawing comparisons between three methods of removable restoration. The data available on this subject are limited in terms of established clinical trials and reviews and must continue to be developed through further research. Analysis of outcome comparisons in terms of clinical performance serves to enhance the understanding of the behavior of removable dentures. Integrating implant treatment into certain challenging clinical cases may improve long-term performance and patient satisfaction.

### 4.5. Strength/Contributions

With recent data indicating the viability of implant-assisted removable partial dentures, this systematic review highlights the potential addition of dental implants as a means to boost parameters of longevity as well as prosthesis performance. While the individual use of tooth and implant abutments continues to show favorable results, the combined use of both may be favored as an alternative in carefully selected clinical cases.

### 4.6. Limitations of the Study

Limitations must be acknowledged, including small sample sizes, short follow-up periods (in certain cases), and non-randomized, highly selective patient cohorts.

There was a high heterogeneity of data within the present systematic review. This stems from a series of factors, the main one being the lack of standardized research protocols among clinical trials. This may be attributed to causes such as variations in systemic parameters such as patients’ ages, sex, and health status as well as oral-health related elements such as types of edentulism, prosthesis design (as well as its components), opposing dentition, or implant placement. Additionally, not all patients within the same cohort participated in the follow-up program, and thus, the case development was not monitored fully in each case. This suggests that while the outcomes are promising, they are not yet generalizable. Still, individual studies contribute valuable insights on the viability of IARPDs in clinical use: improved masticatory performance and patient-reported quality of life (Nogawa et al. [14]), enhanced outcomes with double-crown systems (Zafiropoulos et al. [16]), and optimal implant placement in the anterior mandible (Yi et al. [15]). These aspects emphasize the fact that the presented data must be analyzed with caution and carefully adapted with regard to clinical relevance and applicability.

This study presents a generalized descriptive assessment of the clinical performance of removable prostheses, without analysis of potential confounding factors. Parameters such as prosthesis design (e.g., clasp vs. attachment-retained), implant number, and positioning (anterior vs. posterior regions) and patient-related variables such as bone quality, oral hygiene, and systemic health conditions (e.g., diabetes or osteoporosis) were not reported within this systematic review. These variables can significantly influence clinical outcomes including survival and complication rates. As such, differences in reported performance may not solely reflect prosthesis type but also underlying variations in case selection and clinical protocols.

Another important limitation was represented by the lack of available data on success rates. Clinical studies as well as the other systematic reviews available do not reach a consensus regarding the definition of this term. Certain studies define success as the lack of severe complications presented in each case (as was the criteria for this study), while others consider it to be the absence of any need for reintervention. Thus, the results that are drawn cannot be sufficiently conclusive based on this parameter.

### 4.7. Future Work/Opportunities for Research

While each treatment modality presents specific limitations, all demonstrate clinical value when appropriately indicated. Future research with longer follow-up periods and broader patient inclusion criteria (including analysis based on patient cofactors and prosthesis design) is essential to strengthen the evidence base and guide more universally applicable treatment recommendations, particularly on the usage of implant-assisted removable partial dentures. Future research should incorporate stratified analyses or multivariate modeling to isolate the effect of these confounders more effectively. Additionally, the development of future studies that present a clear definition of the concept of success is an essential element for the better understanding of treatment viability, clinical performance and longevity.

## 5. Conclusions

Based on the data presented within this systematic review, the following conclusions may be drawn:IARPDs demonstrated favorable survival and complication outcomes compared to conventional RPDs;Implant-supported overdentures (IODs) generally achieved the highest survival rates, especially over mid-term follow-up periods;The evidence base for IARPDs remains limited by non-randomized designs, inconsistent reporting of confounders, and short observation periods;Implant-assisted removable prostheses may represent a conservative treatment option in carefully selected patients, although more thorough data are needed for broader application.

However, the evidence remains limited by small sample sizes, heterogeneity, and short follow-up periods. High-quality randomized studies with longer follow-ups and standardized success criteria are needed as well as further investigation into patient-centered outcomes and prosthesis design variables, which will help clarify the role of IARPDs in modern prosthodontics.

## Figures and Tables

**Figure 1 dentistry-13-00389-f001:**
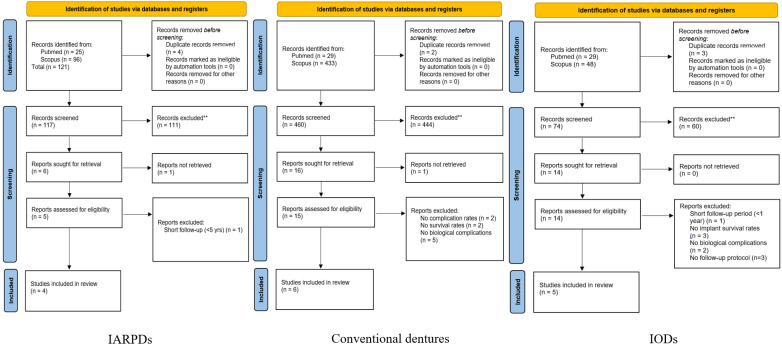
Selection process presented through PRISMA 2020 Flow Diagram [8]. ** All records were excluded manually by 2 researchers.

**Figure 2 dentistry-13-00389-f002:**
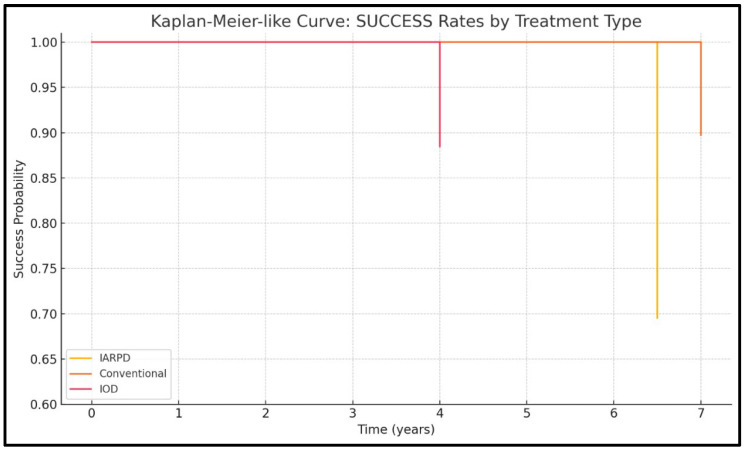
KaplanMeier curve for success rates.

**Figure 3 dentistry-13-00389-f003:**
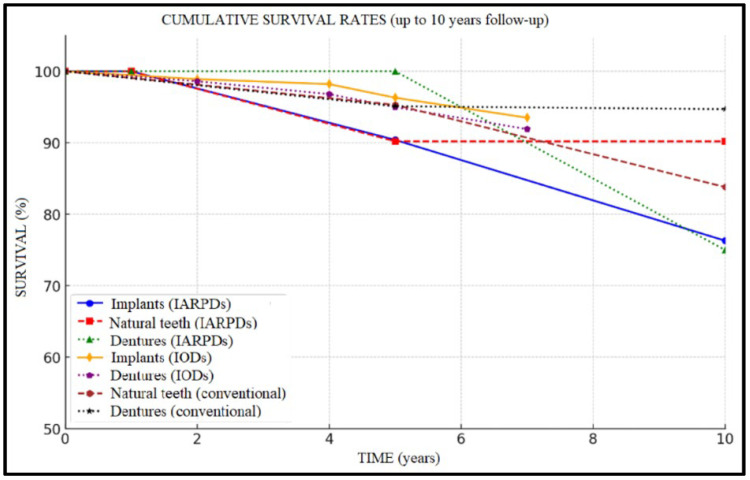
Summary chart on cumulative survival rates (at up to 10-year follow-up).

**Figure 4 dentistry-13-00389-f004:**
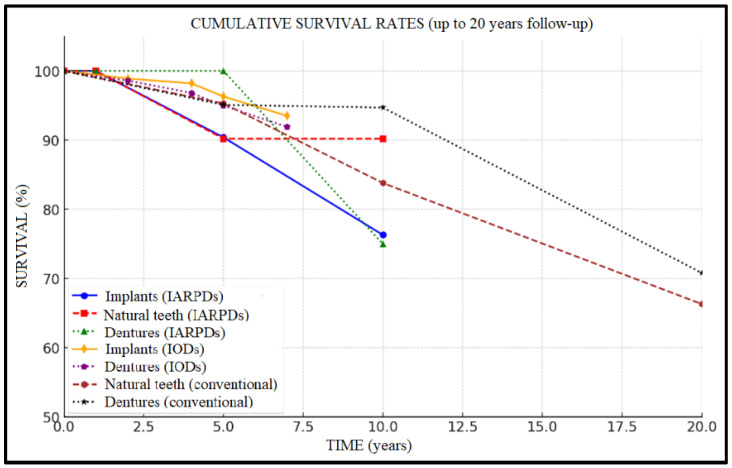
Summary chart on cumulative survival rates (at up to 20-year follow-up).

**Figure 5 dentistry-13-00389-f005:**
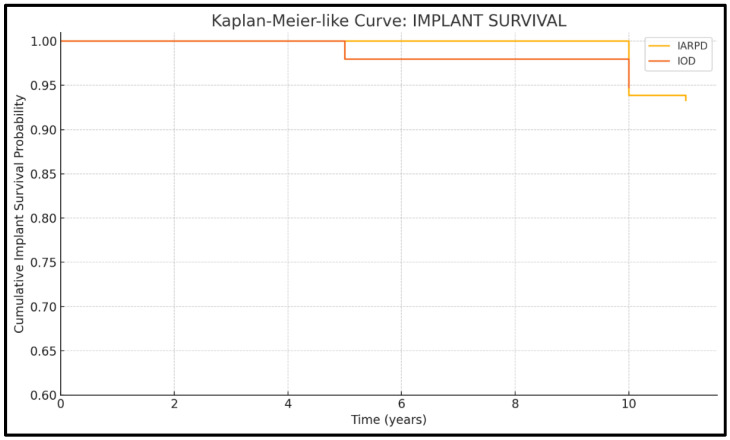
Kaplan-Meier curve for implant survival.

**Figure 6 dentistry-13-00389-f006:**
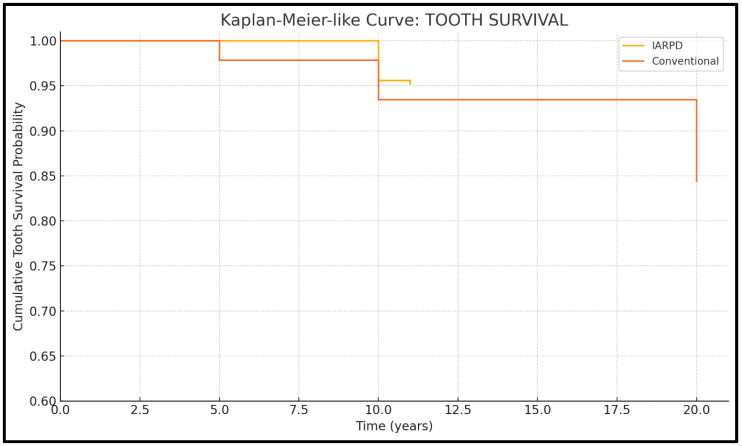
KaplanMeier curve for tooth survival.

**Figure 7 dentistry-13-00389-f007:**
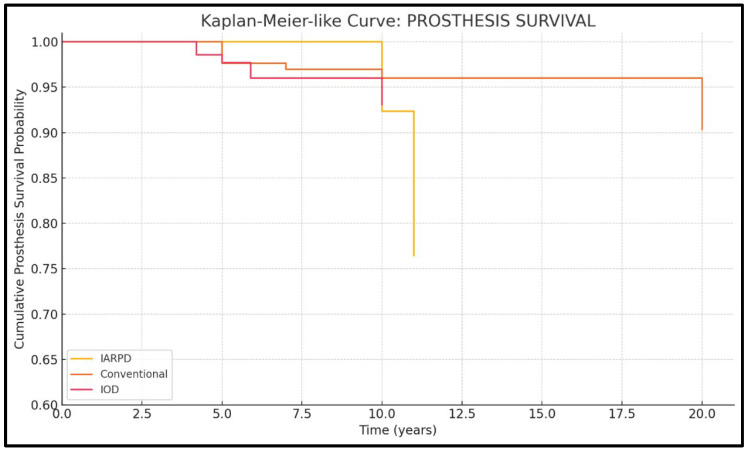
Kaplan-Meier curve for prosthesis survival.

**Figure 8 dentistry-13-00389-f008:**
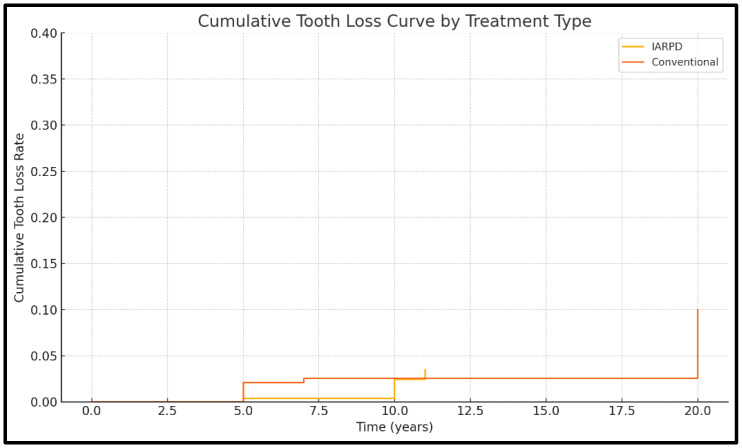
KaplanMeier curve for tooth loss rate.

**Figure 9 dentistry-13-00389-f009:**
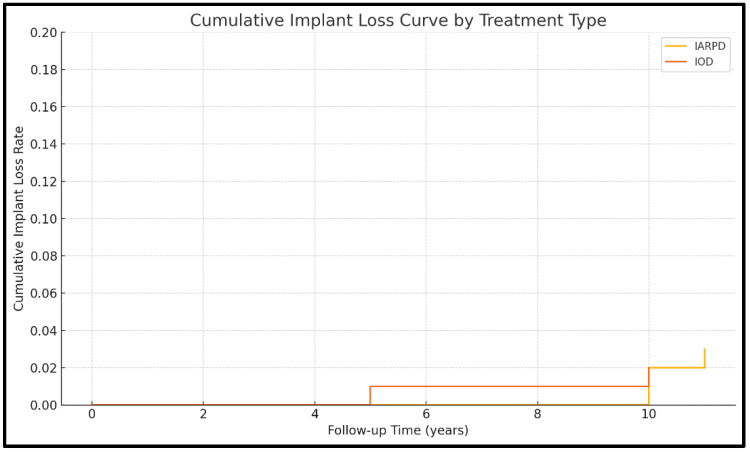
KaplanMeier curve for implant loss rate.

**Figure 10 dentistry-13-00389-f010:**
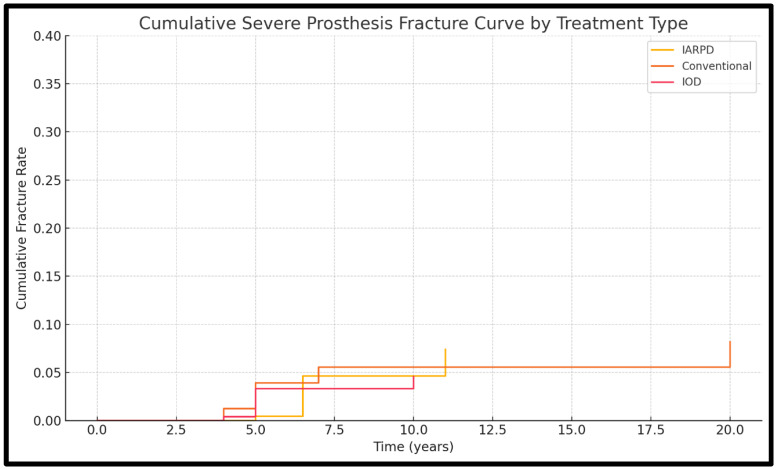
Kaplan-Meier curve for cases of severe prosthesis fracture.

**Table 1 dentistry-13-00389-t001:** Characteristics of the included articles and success and survival rates.

Author	Abutment Type	Prosthesis Type	Cohort Size	Success Rates	Survival Rates	Follow-Up Period	Follow-Up Protocol
Nogawa et al. (2022) [14]	IARPDs	Cast-clasp	4 patients, 4 dentures	-	100% for implants 100% for dentures	1–5 years	Yes
Yi et al. (2023) [15]	IARPDs	Cast-clasp	95 patients, 102 dentures	70.9% for implants 66.7% for dentures	-	~5 years	No
Zafiropoulos et al. (2023) [16]	IARPDs	Double-crown retained	110 patients, 110 dentures	-	99.3% for teeth 99.3% for implants 77% for dentures	~11 years	Yes
Zierden et al. (2024) [17]	IARPDs	Double-crown retained	44 patients, 47 dentures	-	90.2% for teeth 76.3% for implants 75% for dentures	10 years	Yes
Al-Imam et al. (2016) [18]	Conventional	Cast-clasp	65 patients, 83 dentures	-	-	1–5 years	Yes
Jorge et al. (2012) [19]	Conventional	Cast-clasp	53 patients, 53 dentures	-	-	5 years	Yes
Schwindling et al. (2014) [20]	Conventional	Double-crown retained	86 patients, 117 dentures	90% for dentures	93.8% for dentures	7 years	Yes
Stegelmann et al. (2012) [21]	Conventional	Cast-clasp or rod attachment	203 patients, 203 dentures	100% for cast-clasp dentures	-	~28–49 months	No
Wöstmann et al. (2007) [22]	Conventional	Double-crown retained	463 patients, 554 dentures	-	95.3% for teeth 95.1% for dentures	5 years	Yes
Yoshino et al. (2020) [23]	Conventional	Double-crown overdentures	174 patients, 213 dentures	-	83.8% -> 66.3% for teeth 94.7% ->70.8% for dentures	10 years, then 20 years	Yes
Abdoel et al. (2021) [24]	IODs	Bar attachment	133 patients, 133 dentures	-	97.8–100% for implants	5 years	Yes
Ciftci et al. (2023) [25]	IODs	Bar attachment	73 patients, 73 dentures	-	93.5% for implants 91.9% for dentures	7 years	Yes
Klotz et al. (2021) [26]	IODs	Double-crown overdentures	126 patients, 139 dentures	88.2% for dentures	96.2% for dentures	~4.2 years	Yes
Onclin et al. (2023) [27]	IODs	Bar or locator attachment	50 patients, 50 dentures	-	89.5–96.3% for implants 91.3–95% for dentures	5 years	Yes
Zierden et al. (2022) [28]	IODs	Double-crown overdentures	86 patients, 86 dentures	-	92.8% for implants 88.6% for dentures	up 10 years (~5.67 years)	Yes

**Table 2 dentistry-13-00389-t002:** Sample sizes and severe complication occurrences over total follow-up periods.

Author	Prosthesis Type	Total Patients	Abutment Teeth	Abutment Implants	Prostheses	Abutment Loss	Prosthesis Failure	Total Follow-Up
Nogawa et al. (2022) [14]	IARPD	4	16	4	4	0 events	0 events	5 years
Yi et al. (2023) [15]	IARPD	95	N/A	223	102	2 implant loss	N/A	6.5 years
Zafiropoulos et al. (2023) [16]	IARPD	110	153	508	110	2 tooth loss 3 implant loss	25 events	11 years
Zierden et al. (2024) [17]	IARPD	44	120	177	47	6 tooth loss 16 implant loss	3 events	10 years
Al-Imam et al. (2016) [18]	Conventional denture	65	N/A *	X	83	N/A	4 events	5 years
Jorge et al. (2012) [19]	Conventional denture	53	N/A	X	53	N/A	3 events	5 years
Schwindling et al. (2014) [20]	Conventional denture	86	385	X	117	29 tooth loss	6 events	7 years
Stegelmann et al. (2012) [21]	Conventional denture	203	N/A	X	203	N/A	1 event	4 years
Wöstmann et al. (2007) [22]	Conventional denture	463	1758	X	554	66 tooth loss	26 events	8 years
Yoshino et al. (2020) [23]	Conventional denture	174	1030	X	213	235 tooth loss (20-years follow-up)	32 events (20-years follow-up)	10 years, then 20 years
Abdoel et al. (2021) [24]	IOD	133	X	368	133	1 implant loss	8 events	5 years
Ciftci et al. (2023) [25]	IOD	73	X	146	73	3 implant loss	11 events	5.9 years
Klotz et al. (2021) [26]	IOD	126	X	213	139	N/A	2 events	4.2 years
Onclin et al. (2023) [27]	IOD	50	X	200	50	5 implant loss	1 event	5 years
Zierden et al. (2022) [28]	IOD	86	X	465	86	11 implant loss	6 events	10 years

* N/A = data unavailable within the study.

**Table 3 dentistry-13-00389-t003:** Severe and other frequently observed complications and their rates of occurrence.

Author	Abutment Tooth Loss	Implant Loss	Severe Prosthesis Fracture	Other Frequent Complications	Follow-Up Period
Nogawa et al. (2022) [14]	1 case (at 19 months)	-	1 case (at 20 months)	-	1–5 years
Yi et al. (2023) [15]	-	0.4%	8.8%	Perimucositis -> 18.4%, Periimplantitis -> 9.4% Tooth afflictions -> 5.9% Loss of implant retention -> 11.7%	~5 years
Zafiropoulos et al. (2023) [16]	1.88%	1.88%	5.66%	Recessions -> 87% Periimplantitis without implant loss -> 3.77% Ceramic veneer fracture -> 9.43%	~11 years
Zierden et al. (2024) [17]	4.67%	9.04%	-	-	10 years
Al-Imam et al. (2016) [18]	-	-	4 cases (1 base, 3 clasps)	Stomatitis caused by prosthesis -> 15 cases Oral candidosis -> 2 cases	1–5 years
Jorge et al. (2012) [19]	4–7%	-	3 cases (clasps)	Increased tooth mobility -> 21–30% Tooth fracture -> 11–15% Caries development -> 44–46% Acrylic base deformation -> 23–48%	5 years
Schwindling et al. (2014) [20]	3.8%	-	17.1% (6 cases of retreatment)	Abutment tooth treatment -> 15.9% Ceramic veneer fracture/fissure -> 27.7%	7 years
Stegelmann et al. (2012) [21]	1 case	-	7.3% (1 case of retreatment)	Loss of non-abutment teeth -> 5.9% Ceramic veneer fracture/fissure -> 3.9%	~28–49 months
Wöstmann et al. (2007) [22]	66 cases	-	-	Ceramic veneer fracture/fissure -> 26.9% Double-crown decementation ->20.6%	5 years
Yoshino et al. (2020) [23]	3.8%	-	32 cases	Prosthesis remake -> 32 out of 213 dentures due to loss of retention (most frequently) or artificial teeth wear	10 years, then 20 years
Abdoel et al. (2021) [24]	-	1 case	8 cases of retreatment	Within retreatments -> 7 prostheses changed, 1 bar changed Marginal bone loss between −0.40 -> −0.61 mm	5 years
Ciftci et al. (2023) [25]	-	3 implants	6.8–8.2%	Polymer attachment wear -> 72.6% Bar screw loosening -> 10.9%	~5.9 years
Klotz et al. (2021) [26]	-	-	2 cases	Multiple antagonist teeth loss -> 1 case Loss of overdenture retention -> 3 cases	~4.2 years
Onclin et al. (2023) [27]	-	5 implants	1 case	Periimplantitis -> 5.1% (bar attachment)–25.8% (locator attachment)	5 years
Zierden et al. (2022) [28]	-	2.3%	6.9%	Pressure spots -> 70 of 86 cases Acrylic base repair -> 21.2%, Friction loss -> 18%	up 10 years (~5.67 years)

## Data Availability

All data used in the present study are available on the internet, as depicted in the References Section.

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
