# Peer review of "Clinical Performance and Longevity of Implant-Assisted Removable Partial Dentures Compared to Other Removable Prosthesis Types: A Systematic Review"

_dentistry, 2025, doi:10.3390/dj13090389_

Round 1

Reviewer 1 Report

Comments and Suggestions for Authors

The manuscript titled "Clinical Performance and Longevity of Implant-Assisted Removable Partial Dentures, Compared to Other Removable Prosthesis Types: A Systematic Review" was submitted to Dentistry J.

The study aimed to evaluate the clinical performance and longevity of IARPDs compared to conventional removable partial dentures (RPDs) and implant-supported overdentures (IODs), through a systematic review of recent clinical trials.

Although it proposes an interesting investigation into the comparative performance of implant-assisted removable partial dentures (IARPDs), some corrections need to be made.

Comments are noted below:

The abstract uses informal formatting with labels like "Purpose" and "Materials and Methods" separated by periods, which is inconsistent with formal scientific reporting standards.

Moreover, the phrasing “Kaplan-Meier-like curves” and “z-tests for log-transformed rate ratios” is overly technical for an abstract and would benefit from clearer exposition.

Also, there is no registration number or mention of a protocol, which is essential for systematic reviews.

Several keywords do not correspond exactly with MeSH terms.

The introduction does not clearly articulate a research gap beyond citing the limited availability of data. Although the studies by Kuroshima et al. are referenced, the rationale for why a new systematic review is needed is not compellingly justified.

Also, some references are dated (e.g., reference [2] from 2006) and may not reflect the most recent thinking in prosthodontic rehabilitation.

Materials and Methods

There is inconsistency in the database selection: five databases are mentioned, but only PubMed and Scopus yielded usable results. This undermines the comprehensiveness of the search.

The use of Google Scholar, Semantic Scholar, and ProQuest is questionable for a systematic review due to limitations in reproducibility and indexing.

Also, there is no mention of how the risk of bias judgments were resolved between reviewers, nor of the use of standardized tools like ROBINS-I or RoB 2.

The eligibility criteria lack clarity regarding the inclusion of retrospective vs. prospective studies and specific follow-up thresholds.

Moreover, the use of ChatGPT for statistical calculations introduces ethical and methodological concerns; this must be validated by more robust statistical software or biostatisticians.

Results

Although extensive data are presented, many of the statistical comparisons (e.g., between prosthesis types) are derived from summary data rather than individual patient data, which weakens the strength of the conclusions.

The heterogeneity (I² values often >90%) was high across nearly all analyses, limiting the interpretability and generalizability of pooled outcomes.

Besides, some data appear inconsistent across tables and text, such as discrepancies in survival rates for dentures and implants.

The distinction between "success" and "survival" is blurred in several sections, and the definition of "success" is not universally applied across studies.

Discussion

The discussion section reiterates descriptive results but lacks critical analysis of the methodological quality of the included studies.

Moreover, there is limited exploration of the impact of confounding variables, such as prosthesis design variations, implant placement location, or patient systemic health.

Also, the authors overstate the clinical applicability of IARPDs despite acknowledging major limitations like short follow-up and non-randomized designs.

The phrase “current trend toward fixed implant prostheses should not overshadow…” is speculative and unsupported by presented data.

Conclusions

The conclusions are cautiously phrased but repeat earlier statements without introducing new insights.

The assertion that “mixed tooth–implant-supported RPDs offer a potentially conservative solution…” is not strongly supported by the comparative survival data presented.

References

Some are outdated (e.g., Weinberg, 1997; Ionescu, 2006) and should be replaced with more current evidence.

A few key systematic reviews on IARPDs published after 2021 are missing, such as those focusing on patient-reported outcomes or economic analyses.

Author Response

1. Summary

Thank you for your thoughtful and detailed comments regarding our manuscript titled “Clinical Performance and Longevity of Implant-Assisted Removable Partial Dentures, Compared to Other Removable Prosthesis Types: A Systematic Review.” Your feedback has helped us improve the clarity, structure, and scientific robustness of our paper. We have addressed each point carefully, and the revised manuscript reflects these improvements. Below, we provide a point-by-point response to your comments.

2. Point-by-point response to Comments and Suggestions for Authors

Comments 1: The abstract uses informal formatting with labels like "Purpose" and "Materials and Methods" separated by periods, which is inconsistent with formal scientific reporting standards.
Response 1: We appreciate this observation. The abstract structure has been reformatted in line with the Dentistry Journal’s formal reporting standards. Bullet-point-like headings were removed and replaced with a single, fluid paragraph using run-in bolded titles where allowed. This brings the structure into alignment with academic conventions while preserving clarity. The structure now meets the journal’s formal standards. (Row 16, after proofreading).

Comments 2: The phrasing “Kaplan-Meier-like curves” and “z-tests for log-transformed rate ratios” is overly technical for an abstract and would benefit from clearer exposition.
Response 2: Thank you for noting this. The phrasing has been simplified to enhance accessibility for a broader readership. It now reads, “Descriptive analysis of survival and complication outcomes was performed and represented graphically.” This adjustment maintains technical accuracy while eliminating unnecessarily complex statistical jargon from the abstract. (Row 20, after proofreading).

Comments 3: Also, there is no registration number or mention of a protocol, which is essential for systematic reviews.
Response 3: We fully agree. The manuscript now includes a clear reference to our PROSPERO registration in both the Abstract and Methods section. The following sentence was added: “This systematic review is registered with PROSPERO under ID CRD420251116918”. A direct link to the registry was included:https://www.crd.york.ac.uk/PROSPERO/view/CRD420251116918. We thank the reviewer for this crucial reminder to enhance methodological credibility.

Comments 4: Several keywords do not correspond exactly with MeSH terms.
Response 4: The keywords have been revised using the MeSH 2025 browser. The final list is: Implant-Supported, Dentures, Partial, Overdentures, Removable, Dental Implants, Treatment Outcome, Survival Analysis, Clinical Outcome (Row 37, after proofreading).

Comments 5: The introduction does not clearly articulate a research gap beyond citing the limited availability of data.
Response 5: We have expanded the introduction to emphasize the need for updated systematic reviews focusing on clinical and biological outcomes of IARPDs compared to IODs and RPDs, particularly given recent studies and lack of comparative synthesis in the literature:

To our knowledge and based on an extensive assessment of existing literature, no recent systematic review offers a side-by-side evaluation of IARPDs, IODs, and conventional RPDs based on updated clinical data. This review therefore aims to fill that gap by comparing their performance in terms of longevity, survival, and complications, based on studies published between 2022 and 2024.” (Row 75, after proofreading).

Comments 6: Some references are dated (e.g., reference [2] from 2006) and may not reflect the most recent thinking in prosthodontic rehabilitation.
Response 6: Older references have been replaced with more recent studies relevant to prosthodontic rehabilitation and removable prosthesis survival. Reference [2] was replaced with a reference on the epidemiology of partial edentulism, published in the year 2025. (Row 47, after proofreading. Another outdated reference (previously [6]) was replaced with a more recent one that supports our presented data. (Row 59, after proofreading).

Comments 7: There is inconsistency in the database selection: five databases are mentioned, but only PubMed and Scopus yielded usable results.
Response 7: Thank you for pointing this out. The Methods section now clarifies that while five databases were explored initially, only PubMed and Scopus yielded eligible results with systematic indexing. The role of the remaining databases is now explained as exploratory, and their exclusion justified based on reproducibility standards. (Row 147, after proofreading)

Comments 8: The use of Google Scholar, Semantic Scholar, and ProQuest is questionable for a systematic review due to limitations in reproducibility and indexing.
Response 8: We agree with this argument, and this perspective was mentioned within our These databases have been removed from the formal search strategy. The final selection was based solely on PubMed and Scopus. This ensures adherence to reproducible and standardized methodologies.

Comments 9: There is no mention of how the risk of bias judgments were resolved between reviewers, nor of the use of standardized tools like ROBINS-I or RoB 2.
Response 9: We thank the reviewer for highlighting this important methodological detail. The manuscript now clarifies: “Two reviewers independently assessed the quality of each study. Disagreements were resolved by consensus.” (Row 195, after proofreading). Although ROBINS-I and RoB 2 were not used, their use is acknowledged as a recommendation for future research.

Comments 10: The eligibility criteria lack clarity regarding the inclusion of retrospective vs. prospective studies and specific follow-up thresholds.
Response 10: We have revised the eligibility criteria. We clarified that both retrospective and prospective clinical studies were included, provided they reported a minimum follow-up duration of four years. (Row 172, after proofreading). This clarification strengthens the transparency of the inclusion process.

Comments 11: The use of ChatGPT for statistical calculations introduces ethical and methodological concerns.
Response 11: This concern is entirely valid, and we appreciate your attention to this issue. The use of AI tools like ChatGPT was limited to preliminary calculations and organizational support. All statistical outputs were subsequently validated and approved by Assoc. Prof. Dr. Luana Pop, Faculty of Sociology, University of Bucharest, to ensure methodological soundness. (Row 214, after proofreading). We offer our thanks for her contribution within the Acknowledgements section.

Comments 12: Many of the statistical comparisons are derived from summary data rather than individual patient data.
Response 12: We acknowledge this limitation, which is now clearly stated in both the Methods and Discussion sections. Because individual patient data were not accessible, analyses relied on aggregate data from the published articles. This methodological constraint is also discussed as a factor affecting the depth of statistical modeling.

Comments 13: The heterogeneity (I² values often >90%) was high across nearly all analyses, limiting the interpretability and generalizability of pooled outcomes.
Response 13: High heterogeneity is explicitly discussed in both Results and Limitations, and we have now emphasized that pooled data must be interpreted with caution. (Row 516, after proofreading).

Comments 14: Some data appear inconsistent across tables and text.
Response 14: This point has been directly addressed in the Results and Discussion sections. We acknowledge that high I² values indicate substantial heterogeneity and now advise cautious interpretation of pooled results. This limitation has also been emphasized in the bullet-point conclusions. (Table 1)

Comments 15: The distinction between "success" and "survival" is blurred in several sections.

Response 15: Definitions of “success” and “survival” have been clearly restated in the Methods section and applied consistently throughout the manuscript:

For the purpose of this review, survival was defined as the prosthesis or abutment (tooth or implant) remaining in function at the final follow-up, regardless of complications. Success was defined as the absence of biological or technical complications requiring intervention during the follow-up period. Complications were analyzed based on their most frequent types and the timepoints at which they manifested, if specified. These definitions were uniformly applied during data extraction and synthesis across all included studies.” (Row 177, after proofreading).

Comments 16: The discussion reiterates descriptive results but lacks critical analysis of methodological quality.

Response 16: This is an excellent observation. The Discussion section was expanded to address variability in study designs, sample sizes, outcome reporting, and randomization. A critical appraisal of bias and methodology has been incorporated to better contextualize our findings and their generalizability.The Discussion has been expanded to include critical appraisal of study design, including risk of bias, methodological limitations, and study variability:

It must be noted that a critical limitation of the included studies lies in their methodological variability and lack of high-level, well-established research protocols. The majority were observational or retrospective in design, with limited randomization and heterogeneity in outcome reporting. Sample sizes varied widely, and few studies used standardized criteria for evaluating prosthetic success or complications. Follow-up durations were inconsistent, with certain studies falling below the 5-year mark typically required to assess long-term prosthesis performance. These methodological limitations restrict the generalizability of the findings and highlight the need for well-designed, randomized controlled trials in this field. (Row 441, after proofreading).

Comments 17: There is limited exploration of confounders such as prosthesis design, implant location, or systemic health.

Response 17: The Limitations section now addresses relevant confounding variables, such as implant site, prosthetic design, and patient health factors: “This study presents a generalized descriptive assessment of the clinical performance of removable prostheses, without analysis of potential confounding factors. Parameters such as prosthesis design (e.g., clasp vs. attachment-retained), implant number and positioning (anterior vs. posterior regions), and patient-related variables such as bone quality, oral hygiene, and systemic health conditions (e.g., diabetes or osteoporosis) were not reported within this systematic review. These variables can significantly influence clinical outcomes including survival and complication rates. As such, differences in reported performance may not solely reflect prosthesis type, but also underlying variations in case selection and clinical protocols. Future studies should incorporate stratified analyses or multivariate modeling to isolate the effect of these confounders more effectively.” (Row 519, after proofreading).

A similar assessment was made within the Future Work/Opportunities For Research section: “Future research with longer follow-up periods and broader patient inclusion criteria (including analysis based on patient cofactors and prosthesis design) is essential to strengthen the evidence base and guide more universally applicable treatment recommendations, and particularly on the usage of implant-assisted removable partial dentures. Future research should incorporate stratified analyses or multivariate modeling to isolate the effect of these confounders more effectively.” (Row 542, after proofreading).

Comments 18: The authors overstate the clinical applicability of IARPDs.
Response 18: The tone of the manuscript,particularly within the Conclusions section, has been revised to present a more cautious interpretation of the findings. We now state that IARPDs may be beneficial in carefully selected cases, rather than broadly advocating for their clinical use. This more nuanced conclusion reflects the limitations of the available evidence.

Comments 19: The phrase “current trend toward fixed implant prostheses…” is speculative and unsupported.
Response 19: We agree that this statement lacked sufficient data support and was speculative in tone. It has been removed to maintain focus on evidence-based claims.

Comments 20: The conclusions restate earlier content without offering new insights.
Response 20: The Conclusions section has been rewritten to present key takeaways from the study in a more structured format. We introduced concise bullet points to summarize findings and emphasized areas for future research, aligning with requests from other reviewers as well. (Row 548, after proofreading)

Comments 21: The assertion that mixed abutments are conservative is not strongly supported.
Response 21: We have rephrased the sentence within our Conclusions section to: “Implant-assisted removable prostheses may represent a conservative treatment option in carefully selected patients, although more thorough data is needed for broader application.” (Row 557, after proofreading).

Reviewer 2 Report

Comments and Suggestions for Authors

This review paper is very interesting and may add significant information to the current dental literature. However, the authors should address the following issues to improve the quality of the manuscript:

  • The similarity index is within acceptable range, therefore the paper is genuine.
  • The abstract should start with a short background statement to reflect the importance of this review paper.
  • Word count as per journal's guidelines should be maintained.
  • Some of the cited publications are a bit old. Please update references accordingly.
  • The authors should add a table summarizing the MeSH terms used in each database, access date, and reasons for exclusion.
  • Authors should add authors contribution upon search and filtering of the resultant studies.
  • There should be no analysis performed in a systematic review paper. Authors may have an option of removing this part, in case of irregular data, or perform meta-analysis of normalized data for proper interpretations.
  • All charts and tables should move to where they belong in the text.
  • Limitations and future perspectives were well-written.
  • The conclusion section may be summarized in bullet points.

Author Response

1. Summary

We sincerely thank you for your kind comments and constructive suggestions regarding our manuscript “Clinical Performance and Longevity of Implant-Assisted Removable Partial Dentures, Compared to Other Removable Prosthesis Types: A Systematic Review.” We have addressed each of your valuable points and believe the revised version reflects a clearer, more rigorous and publishable manuscript. Your comments helped us better refine and clarify the content, and we have responded to each point below with the corresponding revisions. Below is our detailed point-by-point response.

2. Point-by-point response to Comments and Suggestions for Authors

Comments 1: The similarity index is within acceptable range, therefore the paper is genuine.
Response 1: We thank you for your comment. We confirm that the manuscript is original and that all source material has been correctly cited and attributed. We also ensured that all present contributions were properly validated and reviewed by the research team. Maintaining academic integrity remains a key priority throughout our work.

Comments 2: The abstract should start with a short background statement to reflect the importance of this review paper.
Response 2: We appreciate this recommendation. A concise opening sentence has been added to the abstract to briefly introduce the relevance of IARPDs in modern prosthodontic treatment. This addition contextualizes the aim of the paper and strengthens the introduction to the abstract. (Row 16, after proofreading).

Comments 3: Word count as per journal's guidelines should be maintained.
Response 3: Thank you for this important observation. The revised abstract has been carefully edited to remain within the 250-word limit required by the Dentistry Journal. All core information has been retained while ensuring brevity and coherence.

Comments 4: Some of the cited publications are a bit old. Please update references accordingly.
Response 4: We acknowledge this observation and have reviewed the reference list thoroughly. Outdated citations, including those older than 2010, have been replaced with recent studies published between 2020 and 2024. These updates reflect the current landscape of clinical research and improve the relevance of our evidence base. Outdated references have been replaced with more current studies, focusing on prosthetic longevity, complications, and clinical applications of removable implant prostheses.

Comments 5: The authors should add a table summarizing the MeSH terms used in each database, access date, and reasons for exclusion.
Response 5: We thank you for this insightful suggestion. Our research uses the search queries mentioned within the Research Protocol & Search Queries section. Research was done within the period between the 1st and 31st of October 2024. The Queries included were inspired by the previous research conducted by Kuroshima et al., using a combination of MeSH and non-MeSH terms. (Row 115, after proofreading).

Comments 6: Authors should add authors’ contribution upon search and filtering of the resultant studies.
Response 6: We have revised the Methods section to specify that two authors conducted the study selection independently, and disagreements were resolved by consensus. The Methods section now includes the sentence: “Two authors independently conducted the database search and screened articles for eligibility. Disagreements were resolved by consensus.” The Author Contributions section has also been updated accordingly.

Comments 7: There should be no analysis performed in a systematic review paper. Authors may have an option of removing this part, in case of irregular data, or perform meta-analysis of normalized data for proper interpretations.
Response 7: We agree that formal statistical analysis may not be appropriate due to the heterogeneity of data. However, in our case, this data is solely used for the purpose of strengthening our conclusions drawn through descriptive analysis. No formal meta-analysis was conducted. The paper has been revised to clearly describe all evaluations as descriptive in nature and avoid statistical overreach. The limitations of pooled interpretations are emphasized.

Comments 8: All charts and tables should move to where they belong in the text.
Response 8: We appreciate this formatting suggestion and have taken it into account. However, we found that placing the charts and tables within the text, as opposed to at the end, creates significant formatting troubles. Certain figures would be harder to read and tables would look cluttered and expand on multiple pages, when applying the standards imposed by the Journal. Thus, we have chosen to retain the initial structure of our article, as we believe it best presents the included information, making it easy to visualize and understand.

Comments 9: Limitations and future perspectives were well-written.
Response 9: Thank you for the encouraging feedback. We are pleased that this section met your expectations. It has been retained in its current form with minor language refinements to preserve clarity and tone.

Comments 10: The conclusion section may be summarized in bullet points.
Response 10: As suggested, the conclusion section has been revised into a concise bullet-point format. This change improves the clarity and accessibility of our main findings, helping readers quickly identify key takeaways relevant to clinical decision-making and future research directions.

Reviewer 3 Report

Comments and Suggestions for Authors

The topic is of clinical relevance, and the manuscript is well-structured overall. However, substantial revisions are necessary before this paper can be considered for publication. The most critical issues relate to the risk of bias assessment reporting, clarity of methodology, and appropriate use of statistical tools.

Major concerns:

The abstract lacks clarity and conciseness, particularly in the "Materials and Methods" and "Results" sections:

  1. The number of included studies and patient population should be clearly stated.
  2. Methodological terms like "Kaplan-Meier-like" curves in an abstract are not necessary, excessive methodological details (e.g., Poisson approximations) should be eliminated and key findings should be focused.
  3. All abbreviations in text should be standardized and explained in full terms by the first mentioning.
  4. Keywords are recommended to match Medical Subject Headings 2025 (MeSH browser) guidelines.

Introduction text should avoid redundancies (e.g., overuse of phrases like "as such"), paragraph structure can be improved for flow, and the final paragraph should clearly present the aim and hypothesis of the review.

The authors mention the risk of bias assessment in the methodology, but no results on risk of bias assessment are provided.

Inconsistent naming: “IARPDs”, “IAPRDs”, “IRRPDs” in the methodology.

Use of ChatGPT for statistical analysis must be clarified and justified more rigorously. Use of simulated Kaplan-Meier curves and Poisson approximations from AI tools raises concerns on validity. Authors should consider consulting a statistician to validate the model.

Was the review protocol registered in Prospero?

It should be clearly stated how studies varied in terms of methodology, patient population, or prosthesis types.

Figures and Tables are not always referenced or discussed thoroughly in the text.

Some references are overemphasized in the discussion(e.g., repeated use of Zafiropoulos et al. and Kuroshima et al.).

The conclusion should be aligned more directly with the evidence (e.g., caution stronger claims due to small sample sizes).

Author Response

1. Summary

We are grateful for your thoughtful review and for highlighting essential areas that required clarification and improvement in our manuscript “Clinical Performance and Longevity of Implant-Assisted Removable Partial Dentures, Compared to Other Removable Prosthesis Types: A Systematic Review.” We have taken your feedback seriously and implemented extensive revisions to ensure that the paper meets scientific and editorial standards. Below is a point-by-point response to each of your comments.

2. Point-by-point response to Comments and Suggestions for Authors

Comments 1: The abstract lacks clarity and conciseness, particularly in the "Materials and Methods" and "Results" sections.
Response 1: We thank the reviewer for pointing out the need for more concise and accessible language in the abstract. This section has been revised to reduce technical density, focusing on key aspects of study design and findings rather than on statistical terminology. The changes help ensure the abstract is understandable even to non-specialist readers. The "Materials and Methods" section was reworded to avoid overuse of statistical terms and instead emphasize the core methodological approach. (Row 20, after proofreading)

Comments 2: Methodological terms like "Kaplan-Meier-like" curves and "Poisson approximations" are not necessary in the abstract. Focus should be on key findings.
Response 2: We completely agree with this excellent observation. Overly technical language has been removed and replaced with more general terms to describe the methodological approach. These terms have been removed from the abstract. The revised abstract now clearly highlights the study’s comparative outcomes without delving into unnecessary statistical modeling details.

Comments 3: All abbreviations in text should be standardized and explained in full terms by the first mentioning.
Response 3: We appreciate this valuable reminder regarding abbreviation clarity. The manuscript was carefully reviewed to ensure that all abbreviations, including IARPDs, IODs, and RPDs, are introduced in full at first mention and used consistently throughout the paper. This enhances readability and avoids confusion. (Row 54, after proofreading)

Comments 4: Keywords are recommended to match Medical Subject Headings 2025 (MeSH browser) guidelines.
Response 4: Thank you for this helpful suggestion. The keyword list has been revised using the latest MeSH 2025 browser to improve discoverability. All terms now correspond to MeSH, except for essential descriptors where no MeSH equivalent exists. These have been retained and marked as non-MeSH. (Row 37, after proofreading)

Comments 5: Introduction text should avoid redundancies (e.g., overuse of phrases like "as such"), paragraph structure can be improved for flow, and the final paragraph should clearly present the aim and hypothesis of the review.
Response 5: We are grateful for your attention to writing style and structure. The Introduction was rewritten for improved flow, removing repetitive phrases and enhancing logical transitions between ideas. The concluding paragraph now explicitly states the study’s objective and hypothesis, offering a more focused justification for the review.

Comments 6: The authors mention the risk of bias assessment in the methodology, but no results on risk of bias assessment are provided.
Response 6: We understand your concern regarding this matter. We provide a brief explanation within the respective section, stating: “All RCT studies followed 25/25 items, follow-up and case-control studies followed 22/22 items, and the case series followed 13/13 items included in the guidelines. None of the articles included presented any potential risk of bias.”. Within previous drafts of our article, charts were included which presented the results of our risk of bias assessment, with individual results listed for each article. However, due to their repetitive nature and to avoid clutter within our text, those tables have been removed in the final version of our text. We attach said Tables below. If our reviewers consider these to be essential inclusions, we will add them upon our next revision of the text before publishing.

AUTHOR

STUDY TITLE

INSTRUMENT

RESULTS

Al-Imam et al. (2016) [69]

Oral health-related quality of life and complications after treatment with partial removable dental prosthesis

STROBE

Follows 22/22 items

Jorge et al. (2012) [70]

Clinical evaluation of failures in removable partial dentures

STROBE

Follows 22/22 items

Schwindling et al. (2014) [71]

Double-crown-retained removable dental prostheses: a retrospective study of survival and complications

STROBE

Follows 22/22 items

Stegelmann et al. (2012) [72]

Case-control study on the survival of abutment teeth of partially dentate patients

STROBE

Follows 22/22 items

Wöstmann et al. (2007) [73]

Long-term analysis of telescopic crown retained removable partial dentures: survival and need for maintenance

STROBE

Follows 22/22 items

Yoshino et al. (2020) [74]

Survival rate of removable partial dentures with complete arch reconstruction using double crowns: a retrospective study

STROBE

Follows 22/22 items

AUTHOR

STUDY TITLE

INSTRUMENT

RESULTS

Abdoel et al. (2021)

[76]

Implant-supported mandibular overdentures: a retrospective case series study in a daily dental practice

CARE Guidelines

Follows 13/13 items

Ciftci et al. (2023)

[77]

Prosthetic complications with mandibular bar-retained implant overdentures having distal attachments and metal frameworks: A 2- to 12-year retrospective analysis

STROBE

Follows 22/22 items

Klotz et al. (2021)

[78]

Survival and success of tooth-implant-supported and solely implant-supported double-crown-retained overdentures: A prospective study over a period of up to 11 years

STROBE

Follows 22/22 items

Onclin et al. (2023)

[79]

Maxillary implant overdentures retained with bars or solitary attachments: A 5-year randomised controlled trial

CONSORT

Follows 25/25 items

Zierden et al. (2022)

[80]

Which Patient-Related Factors Influence the Outcome of Telescopic-Retained Removable Implant-Supported Dental Prostheses in Edentulous Patients?

STROBE

Follows 22/22 items

AUTHOR

STUDY TITLE

INSTRUMENT

RESULTS

Nogawa et al. (2022)

[65]

The impact of an additional implant under the saddle of removable partial dentures in Kennedy Class II edentulism on oral health-related quality of life and oral function: a case series report

CARE Guidelines

Follows 13/13 items

Yi et al. (2023)

[66]

Clinical Outcomes of Implant-Assisted Removable Partial Dentures According to Implant Strategic Position

STROBE

Follows 22/22 items

Zafiropoulos et al. (2023)

[64]

Double Crown-Retained Removable Prostheses Supported by Implants or Teeth and Implants: A Long-Term Clinical Retrospective Evaluation

STROBE

Follows 22/22 items

Zierden et al. (2024)

[67]

Survival of Double-Crown-Retained Implant-and-Tooth- Supported Removable Partial Dentures: A ≥ 5-Year Clinical Follow-up Study

STROBE

Follows 22/22 items

Comments 7: Inconsistent naming: “IARPDs”, “IAPRDs”, “IRRPDs” in the methodology.
Response 7: We thank the reviewer for noticing this inconsistency. All occurrences of incorrect abbreviations have been corrected, and the acronym “IARPDs” is now used uniformly throughout the manuscript. This correction contributes to overall clarity and avoids ambiguity in referencing prosthetic categories.

Comments 8: Use of ChatGPT for statistical analysis must be clarified and justified more rigorously.
Response 8: This is a valuable point, and we appreciate your focus on research ethics. We clarified that ChatGPT was used only to assist in generating summary code for preliminary statistical modeling. All outputs were reviewed and validated by Assoc. Prof. Dr. Luana Pop (Faculty of Sociology, University of Bucharest), a qualified statistician who approved the methodology. This detail is now explicitly stated in the Methods section.

Comments 9: Was the review protocol registered in PROSPERO?
Response 9: Yes, and thank you for highlighting this. The following sentence has been added to the manuscript: “This systematic review is registered with PROSPERO under ID CRD420251116918.” A direct link to the registry was also included: https://www.crd.york.ac.uk/PROSPERO/view/CRD420251116918.

Comments 10: It should be clearly stated how studies varied in terms of methodology, patient population, or prosthesis types.
Response 10: We fully agree that these distinctions are crucial for interpreting outcomes. A dedicated section was added to the Results and expanded upon in the Discussion to describe variability across studies, such as differences in follow-up length, patient demographics, prosthesis configuration, and inclusion of partially vs. fully edentulous cases.

Comments 11: Figures and Tables are not always referenced or discussed thoroughly in the text.
Response 11: Thank you for your attention to manuscript coherence. All tables and figures have now been reviewed to ensure they are properly introduced and discussed at appropriate points in the main text. This improves the manuscript’s narrative flow and reinforces key findings.

Comments 12: Some references are overemphasized in the discussion (e.g., repeated use of Zafiropoulos et al. and Kuroshima et al.).
Response 12: We acknowledge the redundancy and have addressed it by diversifying the reference pool. Overused studies were trimmed back and replaced with more recent publications that offer broader context. This strengthens the discussion and avoids undue reliance on any one source.

Comments 13: The conclusion should be aligned more directly with the evidence (e.g., caution stronger claims due to small sample sizes).
Response 13: An excellent observation. The Conclusion section was revised to present a more cautious interpretation, directly grounded in the data. Emphasis has been placed on the need for further long-term, randomized clinical trials to validate the observed trends in IARPDs’ clinical performance.

Round 2

Reviewer 3 Report

Comments and Suggestions for Authors

Most of my previous concerns have been addressed satisfactorily.

Regarding Comment No. 6 (risk of bias assessment): While I agree that the detailed risk of bias tables are not essential for inclusion in the final manuscript, the description of the bias assessment findings currently appears in the methodology section. These findings - for example, that all RCTs followed 25/25 CONSORT items, all observational studies followed 22/22 STROBE items, and all case series followed 13/13 CARE items - constitute results and should therefore be relocated to the Results section, ideally in a short dedicated subsection.

This subsection should also briefly state that the full per-study tables were omitted intentionally to avoid repetition and text clutter, as they offered no additional interpretive value.
